# Impact of Somatic DNA Repair Mutations on the Clinical Outcomes of Bone Metastases from Castration-Resistant Prostate Cancer

**DOI:** 10.3390/ijms241512436

**Published:** 2023-08-04

**Authors:** Maria Concetta Cursano, Emilio Francesco Giunta, Emanuela Scarpi, Chiara Casadei, Alessandra Virga, Paola Ulivi, Sara Bleve, Nicole Brighi, Giorgia Ravaglia, Francesco Pantano, Vincenza Conteduca, Daniele Santini, Ugo De Giorgi

**Affiliations:** 1Department of Medical Oncology, Istituto Romagnolo per lo Studio dei Tumori “Dino Amadori”—IRST S.r.l., 47014 Meldola, Italy; emilio.giunta@irst.emr.it (E.F.G.); chiara.casadei@irst.emr.it (C.C.); sara.bleve@irst.emr.it (S.B.); nicole.brighi@irst.emr.it (N.B.); ugo.degiorgi@irst.emr.it (U.D.G.); 2Unit of Biostatistics and Clinical Trials, Department of Medical Oncology, Istituto Romagnolo per lo Studio dei Tumori “Dino Amadori”—IRST S.r.l., 47014 Meldola, Italy; emanuela.scarpi@irst.emr.it (E.S.); giorgia.ravaglia@irst.emr.it (G.R.); 3Biosciences Laboratory, Istituto Romagnolo per lo Studio dei Tumori “Dino Amadori”—IRST S.r.l., 47014 Meldola, Italy; alessandra.virga@irst.emr.it (A.V.); paola.ulivi@irst.emr.it (P.U.); 4Medical Oncology Department, Campus Bio-Medico University of Rome, 00128 Rome, Italy; f.pantano@policlinicocampus.it; 5Unit of Medical Oncology and Biomolecular Therapy, Department of Medical and Surgical Sciences, Policlinico Riuniti, University of Foggia, 71122 Foggia, Italy; vincenza.conteduca@unifg.it; 6Medical Oncology Department, “La Sapienza” University of Rome, 00185 Rome, Italy; daniele.santini@uniroma1.it

**Keywords:** mCRPC, prostate cancer, DNA damage repair, DDR deficiency, bone metastases, liquid biopsy, sheletal related events, SRE

## Abstract

Up to 80% of castration-resistant prostate cancer (CRPC) patients develop bone metastases during the natural history of disease and about 25% harbor mutations in DNA damage repair (DDR) genes. This retrospective observational study evaluated the prevalence of DDR alterations in CRPC patients and their effect on the clinical outcomes associated with bone metastases. The mutational status of CRPC patients was analyzed per FoundationOne^®^ analysis in tissue biopsy or, when it was not possible, in liquid biopsy performed at the onset of metastatic CRPC (mCRPC). The impact of DDR gene mutations on bone-related efficacy endpoints was evaluated at the time of mCRPC diagnoses. In total, 121 mCRPC patients with bone metastases were included: 38 patients had mutations in at least one DDR gene, the remaining 83 ones had a non-mutated DDR status. DDR mutated status was associated with bone metastases volume (*p* = 0.006), but did not affect SRE (skeletal-related events) incidence and time to SRE onset. Liquid and tissue biopsies were both available for 61 patients with no statistically significant difference in terms of incidence and type of molecular DDR alterations. Mutated DDR status was associated with higher bone metastasic volume, although a not detrimental effect on the other bone-related efficacy endpoints was observed.

## 1. Introduction

In prostate cancer (PCa), comprehensive genomic profiling (CGP) is attractive due to the diversity of emerging treatment options. In particular, defects in Breast Related Cancer Antigens (*BRCA*) *1* and *BRCA2* genes result in impairments of DNA damage repair (DDR) deficiency [1]. DDR-deficient cells are sensitive to the inhibition of poly (ADP-ribose) polymerase (PARP), which results in irreversible DNA damage, cell cycle arrest and cell death. Recently, PARP-inhibitors have been approved for treatment of metastatic castration resistant prostate cancer (mCRPC) patients with pathogenic mutation of BRCA1/2 [2,3,4,5,6]. Beyond *BRCA1/2*, deleterious alterations in other genes have been associated with DDR deficiency: *ATM* and *CHEK2* (sensors of DNA damage), *CDK12* (positive regulator of *BRCA* genes), and *PALB2* and *FANCA* (which interact with *BRCA1* and/or *BRCA2* during DNA repair). These defects cause single-strand annealing, non-homologous end joining and, consequently, genomic instability [7,8,9,10,11]. In PCa, the CGP has been previously characterized using tissue biopsies in order to identify mechanisms of resistance to ARSI (androgen receptor selective inhibitors, such as enzalutamide and abiraterone) [1,12,13]. Tissue testing remains the gold standard for CGP, however, the main site of metastases in PCa is bone, which is challenging to sample, analyze and is associated with high-failure rates of DNA sequencing [1,12]. On the other hand, performing CGP on primary tumor samples can be unsatisfying because of often old archival samples and/or lacking or poor material for analysis [1,12]. GCP of plasma cell-free circulating tumor DNA (ctDNA) offers a compelling and minimally invasive complement to tissue testing. Liquid biopsy may overcome the technical difficulties and high-failure rates associated with bone metastasis biopsy with the added value of reflecting the heterogeneity of the metastatic disease [14,15].

The molecular stratification of mCRPC patients remains an unmet clinical need. *BRCA2* mutations have been associated with more aggressive disease and poor clinical outcomes [16,17], but the prognostic implications of other DDR genes are less well established as well as their impact on bone metastases outcomes. PCa bone metastases affect quality of life (QoL) due to the risk of bone pain and the development of “skeletal-related events” (SREs) [18], which is negatively correlated with survival [19]. Pathological bone fractures, hypercalcemia, spinal cord compression, bone surgery and bone radiation therapy are the five events defined as SREs by the Food and Drug Administration (FDA) [20].

This retrospective observational study evaluated the prevalence on liquid and tissue samples of DDR alterations of mCRPC patients at baseline of the first line treatment and their effect on the clinical outcomes of bone metastases.

## 2. Results

### 2.1. Patients: Sample Disposition and Genomic Profile

From January 2021 to September 2022, 150 mCRPC patients were enrolled in the prospective observational study. Plasma samples were obtained from 105 patients before starting first line treatment for mCRPC, whilst tissue samples from the primary tumor were available for 104 patients. Liquid and tissue biopsy were both available for 61 patients. The baseline characteristics of all patients enrolled in the study are described in Table 1. The distribution of molecular alterations found in tissue and liquid biopsy is resumed in Table 2 and graphically described in Figure 1. A total number of 54 mutations in DDR-related genes were found in liquid biopsy, compared to 34 mutations in tissue biopsy of primary tumor. A total of 6 *BRCA1/2* mutated patients have also experienced other mutations in other DDR-related genes. We included in group A all patients with at least one or more pathogenic mutations in DDR-related genes. No statistically significant difference was found in terms incidence of molecular DDR alterations between tissue and liquid biopsy. In patients with both tissue and liquid samples available, the level of agreement between tissue and liquid biopsy is represented in Table 3. Among the 150 patients enrolled, 11 did not experience mCRPC, remaining hormone-sensitive during the observation period. Among the 139 mCRPC patients enrolled, 18 did not develop bone metastases and, consequently, they have been excluded from the final analyses. We therefore included 121 mCRPC patients with bone metastases, divided into group A (“BRCA/DDR mutated”, 38 patients, 31.4%) and group B (not mutated, 83 patients, 68.6%) according to their molecular status. Among the 121 mCRPC patients with bone metastases divided into group A and group B, 45 patients have both liquid and solid samples. Every patient in group B has no mutation in both liquid and solid biopsy.

### 2.2. Clinical Outcomes

Clinical characteristics of these patients are resumed in Table 4. There were no statistically significant differences between Group A and Group B patients in terms of clinical characteristics: age at diagnosis, Gleason score of the primary PCa, stage at diagnosis, type of presentation of mHSPC (high risk and/or high volume according to CHARTEED and LATITUDE criteria [21,22], and type of first-line treatment for mCRPC received.

Then, we evaluated the impact of DDR gene mutations on bone-related efficacy endpoints at the time of mCRPC diagnosis, by dividing patients in two aforementioned molecular groups.

We investigated differences between the two groups in terms of time from bone metastases onset to death, skeletal metastatic tumor burden (sites and number of lesions), skeletal-related events (SREs) incidence, and time to first on-study SRE. The use of antiresorptive agents (bisphosphonates or denosumab) was similar between the two groups.

Concerning bone sites, we divided the patients in those with lesions localized in the axial skeleton only and those with at least one extra-axial lesion; we did not find any difference according to this parameter between group A and B.

SRE were similar in both groups, in terms of incidence but also of median onset time from the diagnosis of bone metastases. Bone pain did not differ between the two groups. (Table 5)

For the number of bone lesions, we adopted two different threshold values, 4 and 10 lesions; interestingly, we found a higher bone metastatic burden in group A than group B, being statistically significant only for the value of 10 lesions as the threshold (*p* = 0.006) (Figure 2).

We also evaluated the median time from bone metastases onset to death, without finding any difference (HR 0.90 CI 0.46–1.78, *p* = 0.763) (Figure 3).

## 3. Discussion

DDR genes are involved in the mechanisms of genetic instability, the repair of DNA aberrations during cell cycle, and the detection and repair of DNA damage, leading to apoptosis of dangerous mutated cells [23]. In this study, we propose testing all PCa patients for sDRR mutations at diagnosis of mCRPC. Somatic determination of molecular alterations on liquid biopsy and/or on the primary site on historic paraffin preparations (if available) was performed.

The incidence of pathogenic variants of somatic mutations in DDR genes among men with mCRPC varied between 11% and 33% [24], significantly higher than in non-metastatic PCa. *BRCA2* mutations were more frequent when compared to other DDR genes (13%), followed by an *ATM* incidence of 7.3% [25]. The incidence of sDDR mutations in our study is in line with literature data; 25.3% of all patients enrolled presented at least a mutation in DDR-related genes and *BRCA2* was the more frequent one, followed by *ATM*.

*BRCA* alterations have been associated with short metastatic-free survival, short cancer-specific survival (CSS) and are predictive of a response to PARP inhibitors and to platinum salts [6,26].

In PCa, Castro E. et al. demonstrated that *BRCA1-2* mutations were more frequently associated with a Gleason score of ≥8, T3/T4 stage, nodal involvement and metastases at diagnosis [27]. The impact of DDR gene alterations and other biomarkers on the clinical outcome of radiometabolic agents in mCRPC is under investigation [28,29], but there are no data about the association between DDR gene alterations and other clinical-biological features rising during the clinical history of mCRPC as PSA/tumor flare, neuroendocrine differentiation, androgen receptor amplification and others [30,31,32,33].

Furthermore, there are no literature data regarding the impact of sDDR genes alterations on bone outcomes of mCRPC patients. In PCa, bone represents the most preferential target site of metastases with an incidence of nearly 75%; autopsy data indicate that the incidence of metastatic bone lesions is 65–75% in PCa patients [34,35].

Bone metastases led to the disruption of normal bone homeostasis as a result of complex interactions between tumor cells, bone marrow cells, and resident bone cells [36].

Prostate cancer cells are attracted to skeletal tissue by chemotactic cytokines, which normally regulate the migration of Hematopoietic Stem Cells (HSCs) into the hematopoietic stem cell niche: in fact, osteoblastic-induced stromal-derivedfactor-1 (SDF-1 or CXCL12) binds the CXCR4 receptor expressed both by HSCs and PCa cells. The competitive binding of SDF-1/CXCR4 of HSCs and PCa cells leads to the formation of the “onco-niche”, in which PCa cells can migrate and then may stay in a quiescent state that can last over years, or can be activated. In the latter case, they can damage physiological bone remodeling by interfering with normal osteoclastic and osteoblastic activity processes through secretion of paracrine factors, such as transforming growth factor β1 (TGF β1), parathroid-hormone-related peptide (PTHrP) and interleukin 6 (IL-6). The result is an aberrant activation of the RANK/RANK ligand (RANKL) pathway and, consequently, abnormal stimulation of bone resorption [37,38].

In PCa bone metastasis, the first step of enhanced osteolysis is followed by strong osteoblastic stimulation, resulting in an excessive abnormal bone apposition. During the natural history of PCa, the development of SREs negatively correlates with survival: pathologic fractures and metastatic spinal cord compressions are associated with a significantly increased risk of death [38,39,40].

This study investigated differences between the DDR gene alterations-carriers and non-carriers in mCRCP patients with bone metastases in terms of time from bone metastases onset to death, skeletal metastatic tumor burden (sites and number of lesions), skeletal-related events (SREs) incidence, and time to first on-study SRE. We hypothesized that the DNA-repair defects in mCRPC may be associated with poor prognoses in terms of bone related outcomes as an indirect consequence of the acquired survival advantages of these tumor cells.

This study does not demonstrate any difference in bone-related outcomes in sDDR genes mutations carriers compared to non-carriers: incidence of SREs, median onset time of the first SRE from the diagnosis of bone metastases as well as bone pain did not differ between the two groups. However, sDDR mutated status showed an association with higher bone tumor burden in mCRPC: these patients had a superior count of bone metastases at the diagnoses of mCRPC compared to patients with normal sDDR status. It should be noticed that the two groups of patients were balanced in terms of use of antiresorptive agents bisphosphonates or denosumab) and first-line treatment performed for mCRPC [41].

In this study, patients were defined as sDDR-gene mutation carriers if one or more mutation in DDR-related genes was detected in the solid and/or liquid biopsy. It is well- known that genomic alterations can be acquired during the progression of the disease as a consequence of the selective pressure of treatments and biological molecular changes occurring during disease progression itself [42]; consequently, a biopsy of a metastatic site represents the ideal approach to identify molecular alterations. The PROfound trial [6], which evaluated 2792 biopsies of mCRPC, showed that DDR genes alterations were present in 28% of all samples, with a similar incidence considering the primary tumor (27%) or metastatic sites (32%). However, somatic determination on a metastatic site, in particular, bone, may be associated with various biases, as well as possible side effects [12]; the PROfound study, for example, pointed out that 30% of biopsy samples may not be of sufficient quality for gene sequencing [42].

The analysis of free circulating tumor DNA (ctDNA) is a promising approach as it may overcome the difficulties that are associated with obtaining tissue; however, there are no solid data that currently allow the reliable use of this test. In our study, the incidence of mutations in DDR-related genes in liquid biopsy (performed at the diagnosis of mCRPC) is numerically higher as compared to solid biopsy (performed on tissue samples of primary tumor); therefore, there was a substantial agreement between solid and liquid biopsy for DDR gene alterations, with different values according to the gene analyzed. For example, mutations of *BRCA1/2*, *PALB 2*, *FANCA* showed an almost perfect agreement; on the other hand, *ATM* showed a moderate agreement between solid and liquid biopsy. According to these data, liquid biopsy could be a valid tool also in PCa with prevalent bone involvement, in which solid biopsy of the metastatic sites is difficult to perform and also in the case of older primary tissues, which are unlikely to be adequate for molecular analysis [43,44]. It is difficult to establish if the higher number of genomic alterations in liquid biopsy is due to a time- and treatment-dependent increase or to a selection of clones harboring those mutations rather than resulting from diagnostic issues such as assay sensitivity; further study is needed to answer to this question.

The retrospective nature of the data analyzed is a limitation of this study. Another limitation is that patients did not perform a solid biopsy of the metastatic site at the time of diagnosis of mCRPC, which may be more comparable to liquid biopsy, reflecting the selective pressure of treatment received from the diagnosis of PCa to the diagnosis of mCRPC. Standard imaging was performed to diagnose mCRPC (CT scan and/or bone scan), however, more sensitive imaging tools as choline PET scan and, more recently, PSMA PET scan, could have an impact on the number of metastatic bone lesions [45,46].

These results should be considered preliminary and further work is needed to determine the relevance of these findings.

## 4. Materials and Methods

### 4.1. Study Design and Aims

The primary aim was to assess the impact of somatic DDR (sDDR) mutations on clinical course of bone metastases in mCRPC patients.

The impact of sDDR mutations on clinical outcomes of bone metastases is defined as: bone metastatic burden, defined as the number and sites (axial only vs. non-axial) of bone metastases at time of mCRPC diagnosis with standard imaging (CT scan and/or bone scan); bone metastases-specific survival, defined as the time from bone metastases onset to death for any cause; prevalence and type of Skeletal-related-events (SREs); time to first on-study SRE, defined as the time from bone metastases onset to first SRE; Bone pain, defined as necessity of opioid use for bone pain at the diagnosis of bone metastases in mCRPC.

As defined by FDA, SREs have been considered: Pathologic bone fractures, hypercalcemia, spinal cord compression, surgery to bone, and radiotherapy to bone.

The analysis of the impact of sDDR mutations have been performed according to 2 groups: patients with mutation in at least one of these gene *ATM/BRCA1/BRCA2/RAD51/PALB2/FANCA/ATM/CDK12/CHECK2* (Group A, all patients “positive” for genomic defects in DDR genes); no DDR carriers (Group B, all patients negative for genomic defects in DDR genes).

The secondary and point of the study is to evaluate the concordance between liquid and tissue biopsy in terms of presence or absence and type of molecular DDR alterations observed.

Patients were retrospectively included for the analysis at the “Department of Medical Oncology, IRCCS Istituto Romagnolo per lo Studio dei Tumori (IRST) Dino Amadori” (Meldola, Italy) from January 2021 to September 2022 and prospectively observed until the data cut-off time on 31 December 2022.

### 4.2. Patients

Patients enrolled were included in the biological observational prospective study IRSTB073 “Biomarker study: the next generation of prostate cancer biomarkers” (Identifier Code: L3P1380). Local Ethical Committee (“IRST Ethical Committee”) approved the IRST B073 single center prospective study. All patients provided written informed consent.

Main inclusion criteria were: histological or cytological confirmed diagnosis of prostate cancer or unequivocal increased of PSA; Patients must have metastatic and/or inoperable disease; Life expectancy of greater than 3 months; ECOG performance status < 2; Age ≥ 18 years; no previous line of treatment for mCRPC; sample tissue of the primary PCa tumor available for NGS analyses and/or blood sample baseline first-line treatment for mCRPC.

### 4.3. Study Procedures and Somatic Variants Analyses

Next generation sequencing using FoundationOne DX1 (Foundation Medicine^®^, Cambridge, MA, USA) was performed on DNA from tumor-biopsy samples obtained at diagnoses of PCa and/or on circulating cell-free DNA from 10-mL blood sample obtained at diagnoses of mCRPC, if available.

Through genomic testing of plasma and/or tumor tissue (archival, if available), patients were screened for the presence of a deleterious somatic alteration in *BRCA1*, *BRCA2*, *ATM*, *CDK12*, *CHEK2*, *FANCA*, *PALB2*, *RAD51*.

We considered DDR to be mutated in all patients with at least one pathogenic mutation (according to the classification of American College of Medical Genetics and Genomics, ACMG) [13], in one of the DDR genes evaluated (*BRCA1/2*, *ATM*, *FANCA*, *CHEK2*, *RAD51*, *PALB2*, and *CDK12*) in solid and/or liquid biopsy. Patients performing NGS analyses on solid sample and liquid biopsy were included in group B if no mutation was detected in both liquid and solid NGS analysis.

If a mutation in one or more DDR-related genes occurred in liquid biopsy and was not detected in solid biopsy (or vice versa), the patient was included in group A.

### 4.4. Statistical Analyses

The description of the cases was carried out through the use of descriptive statistics such as absolute frequencies and percentage frequencies for variables measured on a nominal or ordinal scale, medians, and intervals of variation for variables measured on a continuous scale.

Comparisons of median values of markers within different clinical features were obtained using the nonparametric Wilcoxon test of medians.

Time to skeletal event (SRE) was calculated as the time between the date of bone metastases onset and the date of the first SRE onset for patients who had at least one skeletal event and the difference between the date of bone metastases onset and last follow-up date for patients who did not have any SRE. Events are represented by patients who had at least one SRE.

The curves of the time-dependent variables were determined with the Kaplan–Meier limit product method and the relative comparisons were made according to the log-rank test.

All *p*-values were obtained considering two-tailed tests and statistical analyzes were performed with SAS statistical software, version 9.4.

For each biomarker, concordance was defined as either positive or negative in both tumor and metastasis and discordance was defined as positivity at one site and negativity at the other or vice versa. For each receptor, the discordance rate (DR) was calculated as the proportion of discordant cases with respect to the total number of patients. The two-sided exact binomial 95% confidence interval (95% CI) was estimated.

The relation between the value and the level of agreement was first reported by Landis and Kock [47], with values indicating agreement as follows: 0.00–0.20, slight agreement; 0.21–0.40, fair agreement; 0.41–0.60, moderate agreement; 0.61–0.80, substantial agreement, and 0.81–1.00, almost perfect agreement (perfect agreement = 1.00).

## 5. Conclusions

Mutated DDR status showed an association with higher bone tumor burden in mCRPC. Nevertheless, DDR mutated patients showed neither a higher incidence of SRE or shorter time to the first SRE. In prostate cancer liquid biopsy could be a valid tool for DDR mutational status assessment.

## Figures and Tables

**Figure 1 ijms-24-12436-f001:**
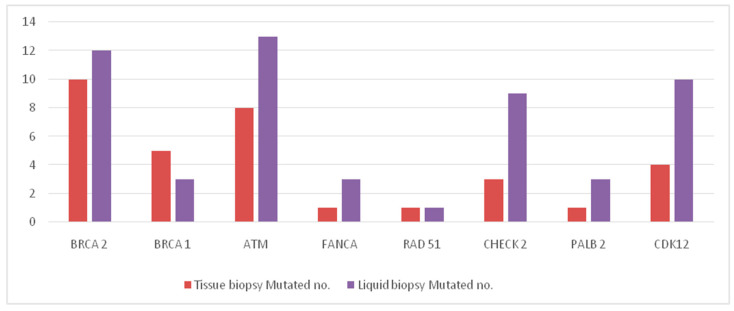
Distribution of molecular alterations found in solid and liquid biopsies.

**Figure 2 ijms-24-12436-f002:**
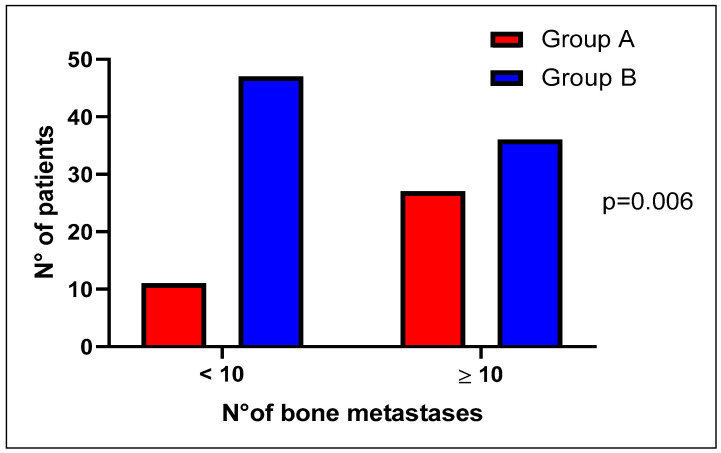
Number of bone metastases in the two molecular groups (DDR mutated, group A versus DDR wild type, Group B), adopting the threshold of 10 lesions.

**Figure 3 ijms-24-12436-f003:**
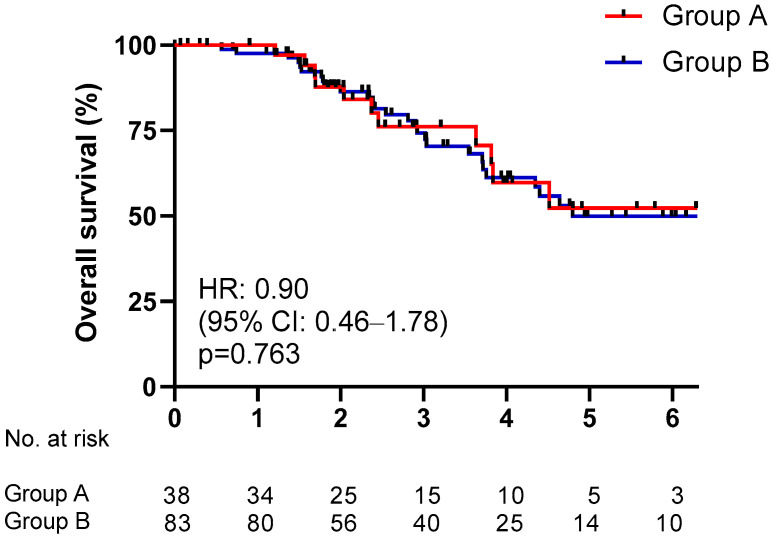
Time from bone metastases onset to death according to molecular status (DDR mutated, group A versus DDR wild type, Group B).

**Table 1 ijms-24-12436-t001:** Baseline patients’ characteristics.

Variable	Number (%) Total = 150
Age at diagnosis, years Median value (range):	65.5 (42–86)
Gleason score	
<8	47 (31.3%)
≥8	92 (61.4%)
Unknown	11 (7.3%)
Stage at diagnosis	
Localized PCa	73 (48.8%)
mHSPC	64 (51.2%)
Type of mHSPC	
High risk and/or high volume	59 (39.3%)
No high risk/volume	66 (44%)
No mHSPC	25 (16.7%)
Sites of metastases in mCRPC	
Bone only	53 (35.4%)
Bone and visceral	11 (7.3%)
Bone and nodes	55 (36.6%)
Bone, visceral and nodes	2 (1.4%)
Visceral or nodes only	18 (12%)
No mCRPC	11 (7.3%)
Treatment for mHSPC	
LhRH analogue	69 (46%)
LhRH analogue + docetaxel	31 (20.7%)
LhRH analogue + ARSI	25 (16.6%)
No mHSPC	25 (16.7%)
Tissue biopsy	
No	46 (30.6%)
Yes	104 (69.3%)
Liquid biopsy	
No	45 (30%)
Yes	105 (70%)
First line treatment for mCRPC	
Chemotherapy	13 (8.7%)
ARSI	126 (91.3%)

Abbreviations: PCa: prostate cancer; mHSPC: Metastatic hormone sensitive prostate cancer; mCRPC: Metastatic castration resistant prostate cancer; LhRH: Luteinizing hormone-releasing hormone; ARSI: androgen receptor selective inhibitor.

**Table 2 ijms-24-12436-t002:** Distribution of molecular alterations found in tissue and liquid biopsy.

Gene	Tissue Biopsy (*n* = 104)		Liquid Biopsy (*n* = 105)	
	Wild Type No.	Mutated No.	Wild Type No.	Mutated No.
*BRCA 2*	94	10	93	12
*BRCA 1*	99	5	102	3
*ATM*	96	8	92	13
*FANCA*	103	1	102	3
*RAD 51*	103	1	104	1
*CHECK 2*	100	3	96	9
*PALB 2*	103	1	102	3
*CDK12*	100	4	95	10
Total mutations	-	33	-	54

**Table 3 ijms-24-12436-t003:** Concordance analysis between solid and liquid biopsy.

Liquid Biopsy				
Tissue Biopsy	WT	Mutated	Total	K Value (95% CI)
	N (%)	N (%)	N (%)	
*BRCA2*				
WT	52 (98.1)	1 (1.9)	53 (100)	
Mutated	0	8 (100)	8 (100)	0.94 (0.80 to 1.00)
Total	52	9	61	
DR (95% CI)	1.64% (0–4.83)			
*ATM*				
WT	51 (96.2)	2 (3.8)	53	
Mutated	1 (12.5)	7 (87.5)	8	0.80 (0.57 to 1.00)
Total	52	9	61	
DR (95% CI)	4.92% (0–10.34)			
*FANCA*				
WT	60 (100)	0	60 (100)	
Mutated	0	1 (100)	1 (100)	1.00 (1.00 to 1.00)
Total	60	1	61	
DR (95% CI)	0%			
*RAD 51*				
WT	60 (100)	0	60 (100)	
Mutated	1 (100)	0	1 (100)	-
Total	60	0	61	
DR (95% CI)	1.64% (0–4.83)			
*BRCA1*				
WT	56 (100)	0	56 (100)	
Mutated	2 (40.0)	3 (60.0)	5 (100)	0.73 (0.38 to 1.00)
Total	58	3	61	
DR (95% CI)	3.28% (0–7.97)			
*CHECK 2*				
WT	56 (96.7)	2 (3.3)	58 (100)	
Mutated	0	3 (100)	3 (100)	0.73 (0.38 to 1.00)
Total	56	5	61	
DR (95% CI)	3.28% (0–7.97)			
*PALB2*				
WT	60 (100)	0	60 (100)	
Mutated	0	1 (100)	1 (100)	-
Total	60	1	61	
DR (95% CI)	0%			
*CDK12*				
WT	56 (98.3)	1 (1.7)	57 (100)	
Mutated	1 (25.0)	3 (75.0)	4 (100)	-
Total	57	4	61	
DR (95% CI)	3.28% (0–7.97)			

Abbreviations: WT: wild type; DR (95% CI): Discordance Rate (95% Confidence Interval).

**Table 4 ijms-24-12436-t004:** Clinical characteristics of mCRPC patients with bone metastasis.

Variable	Total (No. 121)	Group A DDR Mutated (No. 38)	Group B DDR Normal (No. 83)	*p* Value
	No. (%)	No. (%)	No. (%)	
		Age at diagnosis (years)		
Median value (range)	65 (42–85)	65 (52–85)	65 (42–82)	0.773
Gleason score				
<8	41 (33.9%)	14 (36.8%)	27 (32.5%)	0.642
≥8	80 (66.1%)	24 (63.2%)	56 (67.5%)	
Stage at diagnosis				
LocalizedPCa	59 (48.8%)	19 (50%)	40 (48.2%)	0.853
mHSPC	62 (51.2%)	19 (50%)	43 (51.8%)	
mHSPC type				
High risk and/or volume	59 (48.8%)	17 (44.7%)	42 (50.6%)	0.549
No high risk and/or volume	62 (51.2%)	21 (55.3%)	41 (49.4%)	
Sites of metastases (mCRPC)				
Bone only	53	16 (42.1%)	37 (44.6%)	0.772
Bone and visceral	11	4 (10.5%)	7 (8.4%)	
Bone and nodes	55	18 (47.4%)	37 (44.6%)	
Bone, visceral and nodes	2	0	2 (2.4%)	
Type of first line treatment mCRPC				
Chemotherapy	13	4 (10.5%)	9 (10.8%)	0.958
ARSI	108	34 (89.5%)	74 (89.2%)	
Denosumab or bisphosphonates				
Yes	38	10 (26.3%)	28 (33.7%)	0.414
No	83	28 (73.7%)	55 (66.3%)	

Abbreviations: mCRPC: Metastatic castration resistant prostate cancer;ARSI: androgen receptor selective inhibitors.

**Table 5 ijms-24-12436-t005:** Variables in bone metastases positive mCRPC cohort.

Variable	Group A (=38)	Group B (=83)	*p* Value
Age at mCRPC diagnosis (range)	71 (53–86)	69 (44–85)	0.196
Bone sites			
- axial only	10 (26.3%)	33 (39.8%)	
- extra-axial	28 (73.7%)	50 (60.2%)	0.152
Number of bone metastases			
- <4	7 (18.4%)	29 (34.9%)	
- ≥4	31 (81.6%)	54 (65.1%)	0.065
Number of bone metastases			
- <10	11 (28.9%)	47 (56.6%)	
- ≥10	27 (71.1%)	36 (43.4%)	0.006
Incidence of SREs	16/38 (42.1%)	38/83 (45.7%)	0.706
Median time to SRE onset (mo.)	48 (13–not reached)	21 (11–not reached)	0.312
Median time from bone metastases onset to death (mo.)	Not reached	57.6 (44.6–not reached)	0.763
Bone pain			
- No	17 (46%)	45 (57.7%)	0.238
- Yes	20 (54%)	33 (42.3%)	
- Unknown/missing	1	5	

Abbreviations: mCRPC: Metastatic castration resistant prostate cancer; SREs: Skeletal-related events.

## Data Availability

Data is unavailable due to privacy and ethical restrictions.

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
