# Peer review of "Impact of Somatic DNA Repair Mutations on the Clinical Outcomes of Bone Metastases from Castration-Resistant Prostate Cancer"

_ijms, 2023, doi:10.3390/ijms241512436_

Round 1

Reviewer 1 Report

In this study, the authors investigate the impact of DNA damage repair (DDR) gene alterations in CRPC patients with bone metastases and on clinical outcomes associated with bone metastases. The findings are of clinical relevance and provide valuable insights into the relationship between DDR mutations and bone metastasis in CRPC. However, I have some minor concerns that need to be addressed before acceptance for publication.

Minor Concerns:

  • Line 31: The term "SRE" should be defined when first introduced. Please clarify its meaning (skeletal-related events) to avoid any ambiguity for readers.
  • OncoPrint Visualization: it may also be helpful to present the distribution of molecular alterations found in biopsies. Consider incorporating alternative graphical representations to enhance the clarity and completeness of the data presentation.

Author Response

Thanks for your comments and suggestions. We have reported changes in the attached manuscript, following your suggestions, expecially SRE is now fully named  when first introduced and we have introduced an additional figure to graphically represent the data.

Reviewer 2 Report

The Cursano et al. manuscript analzed the impact and prognostic value of DDR gene mutations for bone involvement in mCRPC patients. Despite the results are mainly negative, I think they are of relevance and interest for the community. The study is well described and it is easy to follow the manuscript. However, it is not specified why and how DDR mutations should affect bone metastatic spread. I would encourage to speculate more in the discussion part about and why no differences were found.

Beside this major point I have some minor remarks:

1. Minor grammar/spelling/text issues: line 93; dublicated sentences at line 154 and 156; line 169; 172; 177 either cancer or tumor; line 181: ...onco-niche tumor cells... (what is this, is this a correct nomenclatur?); Line179

2. Abbreviation SRE is not fully named in the abstract.

3. Line 90-96: This description is hard to follow. Maybe add a illustrative scheme of the study groups incl. number of patients and characteristics

4. Figure 2: what is shown on the X-axis. Maybe include in legend the characteristics of group A and B as well e.g. DDR mut vs. WT

5. Please add 1-2 sentences at the end of the discussion or into the conclusion naming the limitation of the study and putative follow-ups.

6. Were any of the included patients treated with radio- or chemotherapy? Here, the response rates should correlate to DDR mutations.

7. line 182: regarding your comment on quiescent or active PCa cells that enter the bone. Isn't it two consecutive steps; 1) the active invasion and met induction and 2) the quiescient stage that can last over years?

8. Line 197: it is not clear why you hypothesized that should be a link between DDR mutations and bone metastastic onset. maybe indirect via aquired survival advantages?

9. Line 222: could you please speculate about DDR alterations in ctDNA at later time points. Is there a time- and treatment dependent increase or selection of clones harbouring those mutations?

10. Line 227: could the observed discrepancy of ATM mutations between local biospy vs. ctDNA arise from diagnostic issues such as assay sensitivity?

Author Response

Thanks for your comments and suggestions. We have reported changes in the attached manuscript, following your suggestions. 

  1. Revised the Minor grammar/spelling/text issues: line 93; dublicated sentences at line 154 and 156; line 169; 172; 177 either cancer or tumor; line 181: ...onco-niche tumor cells... (what is this, is this a correct nomenclatur?); Line179

2. SRE is now fully named in the abstract.

3.  Characteristics of patients included are summarized in Table 1, Table 2 and Figure 1

4.  characteristics of group A and B (DDR mut vs. WT) included in the legend of figures

5. limitations of the study described at the end of the discussion

6.  Yes, but the two groups of patients were balanced in terms of use of antiresorptive agents (bisphosphonates or denosumab) and first line treatment performed for mCRPC (chemotherapy or arsi). 

7. Revised line 182: regarding your comment on quiescent or active PCa cells that enter the bone. Isn't it two consecutive steps; 1) the active invasion and met induction and 2) the quiescient stage that can last over years?

8. Revised Line 197: it is not clear why you hypothesized that should be a link between DDR mutations and bone metastastic onset. maybe indirect via aquired survival advantages?

9. Revised Line 222: could you please speculate about DDR alterations in ctDNA at later time points. Is there a time- and treatment dependent increase or selection of clones harbouring those mutations?

10. Revised Line 227: could the observed discrepancy of ATM mutations between local biospy vs. ctDNA arise from diagnostic issues such as assay sensitivity?
